# Effectiveness of Music Therapy in People Living with Dementia: An Umbrella Review Protocol

**DOI:** 10.3390/ijerph20043200

**Published:** 2023-02-11

**Authors:** Luís Sousa, Cláudia Oliveira, Margarida Tomás, Maria do Céu Pires, António Almeida, Helga Oliveira, E. Begoña García-Navarro, Helena José

**Affiliations:** 1School of Health Atlântica (ESSATLA), 2730-036 Oeiras, Portugal; 2Comprehensive Health Research Centre (CHRC), 7000-811 Evora, Portugal; 3Research in Education and Community Intervention, Piaget Agency for Development, 4410-372 Vila Nova de Gaia, Portugal; 4Health Sciences Research Unit: Nursing (UICISA: E), Coimbra Nursing School, 3045-043 Coimbra, Portugal; 5Nursing Research, Innovation and Development Centre of Lisbon (CIDNUR), Nursing School of Lisbon (ESEL), 1600-096 Lisbon, Portugal; 6Campus El Carmen, Universidad de Huelva, Avda. Tres de Marzo, s/n, 21071 Huelva, Spain

**Keywords:** dementia, Alzheimer disease, music therapy, art therapy, systematic review

## Abstract

Background: Dementia stands out as a neurological disorder which constitutes a progressive decline in cognitive, behavioral, emotional, and social functioning. However, non-pharmacotherapy, such as music therapy, can be combined with pharmacological treatment as a possible strategy to improve functionality regarding the cognitive and non-cognitive dimensions of people diagnosed with dementia. Objectives: To analyze and synthesize published evidence regarding the effectiveness of music therapy in people diagnosed with dementia, concerning cognitive and non-cognitive outcomes. Design: Descriptive study protocol of an umbrella review. Methods and analysis: An umbrella review method will guide this study, focusing on an extensive search of published systematic reviews and meta-analyses reviews that include randomized controlled trials and other types of trials. Databases for the article search include ISI Web of Knowledge, Scopus, and Joanna Briggs Institute (JBI) EBP database, and EBSCO Host platform (Cochrane Database of Systematic Reviews, MEDLINE, and CINAHL). Two reviewers will independently review all titles and abstracts and identify articles considering the inclusion criteria. Afterward, two reviewers will independently extract relevant information from each article for the characterization table, and evaluate the quality of selected articles using the Measurement Tool for Evaluating Systematic Reviews (AMSTAR) 2 guideline. Relevance to clinical practice: Data from this study will aid in designing healthcare workers’ training courses, clinical intervention guidelines, and specific intervention protocols that support pharmacological interventions in treating dementia.

## 1. Introduction

Between 1990 and 2016, the number of people affected by dementia has seen a rising global tendency of 117%, probably due to an aging population [1]. Currently, this rise presents an estimated increase of 57.4 to 152.8 million cases from 2019 to 2050 [2]. Additionally, in 2019, 5% of all deaths in the European Union were related to Alzheimer’s and other dementias [3].

However, there has been a growing body of evidence from North America and Europe suggesting a downward trend in the incidence of dementia, potentially being associated with increased education and improved management of cardiovascular disease and its risk factors [2].

Dementia constitutes an insidious syndrome delineated by a declining progression of mental functions [4] involving changes in cognitive, emotional, behavioral, and social relationships [5,6]. Concurrently, approximately 90% of those affected by the disorder experience, at any stage, a range of neuropsychiatric or behavioral and psychological symptoms of dementia (BPSD), including a broad range of non-cognitive symptoms related to five dimensions: perception, mood, behavior, personality, and basic functioning [5,7].

Furthermore, people living with dementia often experience debilitating neuropsychiatric symptoms that negatively affect their quality of life [5,6].

Considering the limited effectiveness of pharmacological approaches for progressive dementia, non-pharmacological approaches come through as a valuable adjunctive approach, and there have been suggestions for a variety of modalities, such as art therapy [5], with an emphasis on music therapy [8,9,10,11].

Despite the current evidence on the arts and dementia presenting several limitations related to the description, explanation, communication, and simplification of artistic interventions, there is a proposal for a taxonomy that fits music as a modality of art therapy [12].

Art therapy interventions, in general, can successfully improve the care, well-being, and quality of life of people with dementia [13,14]. Furthermore, research on arts and dementia still lacks appropriate theoretical frameworks [15], and studies that have attempted to explain how the arts “work” in people with dementia are scarce [16].

In this context, the arts can bring a patient’s identity to life and be a useful tool for providing person-centered care [17], improving the quality of care and representing a vital element for these people [18]. In this sense, art therapy, in general, and music therapy in particular, need a specific approach, taking into account the stage of dementia, preferences, expectations, and culture of both people with dementia and their family caregivers to meet their needs. Thus, the intervention must be evaluated to capture its essence and be personally meaningful and enriching for the people it is being provided for [19].

The effect of music therapy may significantly improve verbal fluency and decrease anxiety, depression, and apathy in this population, albeit results related to cognition or daily functioning show no significant improvement, and ambiguity on quality of life and agitation is apparent [20]. However, other studies show a symptomatic improvement, with a deferring of dementia’s progression, consequently enhancing caregivers’ well-being while decreasing their burden [21].

Music as an art form has benefits for people with dementia. However, therapeutic musical interventions still need to establish an adequately clear understanding of these effects due to the rich variety of community musical activities available [12].

Despite the evident benefits in these systematic reviews about the use of music therapy in people living with dementia, some research questions arise that justify the need to summarize the evidence.

What is the effect of music therapy on cognitive and non-cognitive outcomes in people living with dementia who live in a community or health institutions?What music therapy modalities and defined characteristics should we consider when implementing music therapy programs for people living with dementia and residing in a community or health institutions?What are the facilitating elements and possible barriers encountered when developing and implementing music therapy interventions in people with dementia?

In this sense, this umbrella review will analyze and synthesize the findings of published systematic reviews and meta-analyses on the effectiveness of music therapy in people living with dementia to assess its potential benefits in terms of cognitive and non-cognitive outcomes.

## 2. Materials and Methods

### 2.1. Design

This study consists of an umbrella review of the systematic reviews and meta-analyses published that included randomized controlled trials and other types of trials.

The umbrella review methodology examines the quality and synthesizes the findings of multiple reviews in a field of research [22,23]. This type of research has nine steps: 1. justify the need for revision; 2. design the study and record the review protocol; 3. elaborate the research strategy and conduct the bibliographical research; 4. identify the revisions included; 5. report the methodological quality of the included reviews; 6. present the results of the review; 7. report the findings and summary of evidence; 8. declare the limitations of the review report; 9. conclusions [23], see Table 1.

The protocol is registered in the Open Science Framework (OSF) with the Prospective Register of Systematic Reviews. The registration number from the Open Science Framework is: osf.io/m6quv.

Moreover, the protocol’s writing follows the statement of Preferred Reporting Items for Systematic Reviews and Meta-analysis Protocols (PRISMA-P) [24]. Accordingly, Figure 1 presents the procedures established for this comprehensive review.

### 2.2. Search Strategy

#### 2.2.1. Database Search

A systematic search will be carried out for articles indexed in MEDLINE, CINAHL, Cochrane Database of Systematic Reviews through EBSCOHost, Scopus, ISI Web of Knowledge, and in the JBI EBP database.

#### 2.2.2. Search Terms

The search strategy will be developed in the EBSCOHost and adapted for the remaining databases. To achieve the initially defined objectives, search equations will be used combining the chosen descriptors with the Boolean operators AND and OR (Table 2 and Appendix A). P (Patient)—Dementia* OR Alzheimer*; I (Intervention)—Music* OR Art therapy; S (Study design)—Review* OR meta-analysis.

### 2.3. Selection Criteria

#### 2.3.1. Types of Studies

Works published in English, Portuguese, and Spanish will be considered for inclusion. The design of the studies to include are: systematic reviews that may be randomized clinical trials (RCTs) or other types of studies (not RCTs), such as before–after studies, non-randomized, and quasi-randomized trials.

#### 2.3.2. Types of Participants

This study will focus on adult patients diagnosed with any types of dementia and any stages of dementia. Even though older adults constitute the main bulk of the population stricken by dementia, this study also seeks to include people with early onset dementia, corresponding to people 64 years old or younger. Therefore, we decided to include the adult population affected by dementia in this study [25].

#### 2.3.3. Types of Interventions

Music therapy presents different implementation methods, such as individual or group sessions with active or passive participation, and via several modalities, such as songwriting, music and relaxation exercises, lyrics discussion, listening to directed music, singing/toning, moving to music, video recording and creation, and lessons on adapted instruments [20]. Consequently, studies using any music therapy modality and studies presenting the effects of these modalities fulfill the inclusion criteria. Studies with music therapy modalities combined with other interventions (e.g., cognitive enhancement therapies, meditation, drawing) will be excluded.

#### 2.3.4. Comparator(s)/Control

The control group will present two modes of intervention: active management, meaning routine care and other types of non-pharmacological interventions; and inactive control, meaning waiting list, with no treatment or blank control.

#### 2.3.5. Types of Main and Secondary Outcome

Cognitive, psychological, and behavioral symptoms of dementia will be included as primary study outcomes. Secondary outcomes will include daily functioning, physiological outcomes, and quality of life. The characteristics and conditions for implementing music therapy modalities will also be considered. Additionally, facilitating elements and possible barriers encountered in developing and implementing interventions in this area may be considered.

### 2.4. Screening and Study Selection

According to inclusion criteria, two reviewers will screen and select studies based on the title and abstract. All records will be retrieved and stored in Mendeley^®^ V1.19.8 (Mendeley Ltd., Elsevier, Amsterdam, The Netherlands), and duplicates will be removed. Citations will be imported into Rayyan QCRI (Qatar Computing Research Institute (Data Analytics), Doha, Qatar) for sorting. In addition, bibliographic references of the included articles will also be screened.

The methodological quality assessment of the systematic reviews and meta-analyses that will be included, and two independent reviewers will carry out the data extraction. A third independent reviewer will resolve conflicts and discrepancies in the evaluation between reviewers.

### 2.5. Data Extraction

Two independent reviewers will extract the following data to the table of characteristics: article features (authors, year of publication, language of publication and country/region), participants (sample size and type and stage of dementia), intervention (mode of music therapy, program duration, frequency, and duration of music therapy session for both treatment and control groups), outcome measure (e.g., QoL instrument, depression, anxiety, functional capacity), authors’ original conclusions, setting (home, long-term care, primary health care, nursing home, social institutions).

When data are missing or information is unclear, a contact to the corresponding authors will be made to retrieve critical information.

### 2.6. Quality Assessment

Quality and control for bias will be assured by two reviewers independently. The Assessment of Multiple Systematic Reviews (AMSTAR) guideline tool, second version, will be adopted. The AMSTAR 2 tool consists of 16 items and will assess the following aspects: (1) whether the PICO component was used (Population, Intervention, Comparison, and Outcomes), and checks regarding the population and intervention whether there was a comparison group is described, and what the outcome was; (2) if the study design is described in detail and if there was a deviation from the protocol; (3) if the authors explain how the study designs were selected; (4) whether a comprehensive search strategy was used; (5) how they dealt with duplicate studies; (6) if the authors performed the extraction of duplicate data; (7) if the list of excluded studies is presented; (8) whether the included studies are described in detail; (9) if a satisfactory technique was used to deal with the risk of bias; (10) if the funding sources of the included studies were described; (11) if the authors used an appropriate statistical combination of results to perform meta-analyses; (12) if the authors assessed the potential impact of risk of bias in individual studies; (13) whether the risk of bias in individual studies was included in the discussion of results; (14) whether a satisfactory explanation of the heterogeneity was given; (15) whether a study on publication bias was carried out; and (16) if there were conflicts of interest [26,27].

The 16 items of the AMSTAR 2 tool present the following quality classification scale of the included studies: high, moderate, low, or critically low. According to this tool, the “High” classification means that there are none or that there is only one non-critical weakness; that is, the systematic review provides an accurate and broad summary of the results of available studies on the topic under study. The classification of “Moderate” means that there is more than one weak point but they are not critical; that is, the systematic review has more than one weak point, but it does not have critical flaws. “Low” is assigned to studies that have a critical flaw with or without non-critical weaknesses; that is, the review has a critical flaw and may not provide an accurate and comprehensive summary of available studies on the topic of interest. Studies are rated “Critically Low” when there is more than one critical failure with or without non-critical deficiencies; that is, the review has more than one critical failure and should not be used to provide an accurate and comprehensive summary of available studies [27].

### 2.7. Strategy for Data Synthesis

Whenever possible, a combination of narrative and quantitative methods will be used in order to synthesize data. Information related to the music therapy modality, number of RCTs, type of dementia and stage, number of participants, instruments that allow evaluating outcomes, and adverse events, constitute the parameters of interest. AMSTAR 2 results will be reported in a descriptive and summary form. Data will be synthesized when homogeneous data are available on three or more included studies. However, the data are expected to show a certain amount of heterogeneity, so we will use a random effects model.

### 2.8. Patient and Public Involvement

No patient is involved nor is the general public.

## 3. Results

The results will be presented in the form of tables. If homogeneous results are found, a meta-analysis graph will be displayed. The remaining data will be synthesized narratively. We intend to find contributions to the understanding of music therapy, the types of modalities, under which conditions it was applied, main cognitive and non-cognitive benefits, the facilitating aspects, and the difficulties and strategies in implementing the music therapy.

## 4. Discussion

The umbrella review will provide a high-level methodological synthesis to identify the most effective and safe music therapy modalities for people living with dementia and their family caregivers.

The findings of this review may contribute to designing a more personalized intervention that is more meaningful [18]. Similarly, it may produce relevant contributions to understanding and knowledge of: (1) what artistic elements or materials should be embedded; (2) what artistic focus, activity, approach, or facilitator should be involved; (3) what setting is appropriate for the intervention; and (4) what principles and competencies facilitators must have [12] when applying music therapy.

As secondary results, facilitators and barriers to implementing music therapy can be identified. Some studies have identified using implementation leaders and volunteers, staff involvement, and integrating song selection and playlist development into clinical practice as facilitators. Barriers such as continuous and unexpected funding, the time it takes to prepare playlists, and staff turnover of health teams are demonstrated [28].

Providing a comprehensive view of music therapy modalities with the highest level of evidence can provide the much need impetus to establish these interventions as part of what should be considered usual care. There is still a prevalent lack of health professionals that can execute and assess the implication of such interventions. More investment in training and in recognizing its importance as part of service users’ healthcare plans may be argued with the results of this study.

## 5. Strengths and Limitations of This Study

This is the first protocol for an umbrella review of non-pharmacological interventions that use music therapy modalities in dementia care. Two independent reviewers will evaluate the research results from screening, selection, data extraction, and methodological quality assessment to data synthesis. If there are discrepancies in the evaluations, a third reviewer will be included.

As this is a comprehensive review, the possible limitation will be related to the quality of the results, which depend on the quality and content of the systematic reviews and meta-analyses available.

Another possible issue may be related to the inclusion of articles published in English, Spanish, and Portuguese. Systematic reviews and meta-analyses published in other languages will certainly be lost.

## 6. Conclusions

Evidence from this study on music therapy modalities will provide evidence for clinical intervention guidelines, training plans, programs, and specific intervention protocols co-adjuvating pharmacological interventions in the treatment of dementia.

It will also aid health professionals with decision making in their daily practice, and on how and under what conditions to apply music therapy modalities.

## Figures and Tables

**Figure 1 ijerph-20-03200-f001:**
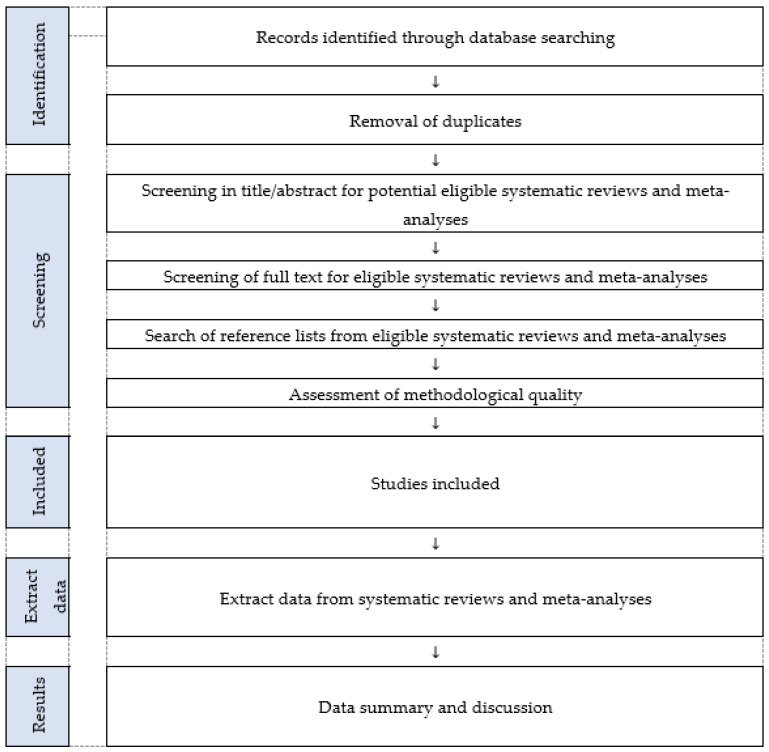
Flowchart of the umbrella review.

**Table 1 ijerph-20-03200-t001:** Nine-step pathway for conduct of an umbrella review [23].

Action Pathway	Description
1. Justify the review	▪ Present a rationale that justifies the relevance of the umbrella review ▪ Produce an argument clarifying knowledge gains entailed by the study ▪ Guarantee evidence appraisal and data extraction by a minimum of two authors
2. Design the study and register the review protocol (optional)	▪ Title and research questions design derive from PICO ▪ State the type of review (umbrella review) in the title ▪ Comply with protocol requirements from chosen registration site or umbrella review guide
3. Set up the search strategy and proceed with the literature search	▪ Include comprehensive review objectives and inclusion criteria that will address the review questions ▪ Define the study’s variables, preferably upon PICO or a variation
4. Determine reviews for inclusion	▪ Produce a flowchart of the search process ▪ Screen studies for inclusion by at least two authors
5. Establish methodological quality	▪ Perform quality assessments by two reviewers independently ▪ Apply appropriate quality checklists
6. Display review results	▪ Describe the included reviews ▪ Summarize the process narratively ▪ Feature significant findings from individual reviews
7. Report findings and evidence summary	▪ Describe/name the included interventions ▪ Inform the synthesized results/outcomes ▪ Summarize results
8. Declare review limitations	▪ Describe limitations and benefits
9. Conclusion	▪ Summarize findings ▪ Set forth implications/further research

PICO = P—Patient, problem or population; I—Intervention; C—Comparison, control or comparator; O—Outcome(s).

**Table 2 ijerph-20-03200-t002:** Search strategy protocol.

Database	Search Strategy
The Joanna Briggs Institute EBP database	((Dementia OR Dementia* OR Alzheimer) AND (Music OR Music* OR Music Therapy OR Art therapy) AND (Review OR Review* OR meta-analysis))
ISI Web of Knowledge	#1: ((TI = (Dementia*)) OR AB = (Dementia*)) OR AK = (Dementia*); #2: ((TI = (Alzheimer*)) OR AB = (Alzheimer*)) OR AK = (Alzheimer*); #3: ((TI = (Music*)) OR AB = (Music*)) OR AK = (Music*); #4: ((TI = (art therap*)) OR AB = (art therap*)) OR AK = (art therap*); #5: ((TI = (Review*)) OR AB = (Review*)) OR AK = (Review*); #6: ((TI = (meta-analysis)) OR AB = (meta-analysis)) OR AK = (meta-analysis); #7: #1 OR #2; #8: #4 OR #3; #9: #6 OR #5; #10: #9 AND #8 AND #7
Interface—EBSCOhost Database—MEDLINE Complete; Nursing & Allied Health Collection: Comprehensive; Cochrane Central Register of Controlled Trials; Cochrane Database of Systematic Reviews; MedicLatina; CINAHL Complete	((Dementia) OR (Dementia*) OR (Alzheimer)) AND ((Music) OR (Music*) OR (Music Therapy) OR (Art therapy)) AND ((Review) OR (Review*) OR (meta-analysis))
Scopus	(dementia OR Alzheimer) AND (music OR music* AND therapy OR art AND therapy) AND (review OR meta-analysis) ((TITLE-ABS-KEY (dementia*)) OR (TITLE-ABS-KEY (alzheimer*))) AND ((TITLE-ABS-KEY (music*)) OR (TITLE-ABS-KEY (art AND therap*))) AND ((TITLE-ABS-KEY (meta-analysis)) OR (TITLE-ABS-KEY (review*)))—1100

## Data Availability

Not applicable.

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
