# Peer review of "Effectiveness of Music Therapy in People Living with Dementia: An Umbrella Review Protocol"

_ijerph, 2023, doi:10.3390/ijerph20043200_

Round 1

Reviewer 1 Report

The protocol is clearly written, I  have the following suggestions for the authors to consider:

1. The authors states"(MD) will be applied when the same scale was used to measure the results." It is not necessary to state you will use MD to synthesize the data, sometimes, a generic inverse variance method might be more common to synthesize the 'homogeneous' data.

2. "The random-effects model will be used to minimize potential heterogeneity when the I2 value is greater than 50%."

You should not base on I^2 to choose a fix-effect model or a random-effects model. Many papers have shown this is not a good practice. You should base on your assumption to choose your model, if you think there is an overall common music effect underline the whole population you should choose a fixed effect model, while if you think there were no such common effect, but effects can be varied for different populations you may choose a random effect model. 

Author Response

Dear reviewer 1

Thank you very much for recognizing our work. Thank you for the opportunity to improve our manuscript with your valuable comments and recommendations.

The protocol is clearly written, I have the following suggestions for the authors to consider:

Thank you very much for recognizing our work.

  1. The authors states"(MD) will be applied when the same scale was used to measure the results." It is not necessary to state you will use MD to synthesize the data, sometimes, a generic inverse variance method might be more common to synthesize the 'homogeneous' data.

Thanks for the suggestion. We will remove that sentence as per your recommendation.

  1. "The random-effects model will be used to minimize potential heterogeneity when the I2 value is greater than 50%."You should not base on I^2 to choose a fix-effect model or a random-effects model. Many papers have shown this is not a good practice. You should base on your assumption to choose your model, if you think there is an overall common music effect underline the whole population you should choose a fixed effect model, while if you think there were no such common effect, but effects can be varied for different populations you may choose a random effect model. 

Thanks for the suggestion. We will clarify according to your recommendation.

We clarify by writing that “However, the data is expected to show a certain amount of heterogeneity, so we will use a random effects model.”

Reviewer 2 Report

This paper focuses on the treatment of dementia, specifically on the use of music therapy and its combination with pharmacological treatment as a possible strategy to improve functionality with respect to cognitive and non-cognitive dimensions of people diagnosed with dementia.   From a study and analysis of the published evidence regarding the effectiveness of music therapy in people diagnosed with dementia, in relation to cognitive and non-cognitive outcomes.

The objective of the work is clear and the methodology and analysis allows to achieve it. However, there are aspects that could be improved:

In the methodology, the  umbrella review protocol is used Include a reference that supports and justifies this approach.

Type of participants. It would be convenient to justify the age range of the study. Why over 18 years old?

For a better global and specific understanding of each activity developed, a diagram could be included that individualizes the macro stages of the methodology and the tasks developed in each one of them.

The results could include a summary of the most relevant results (scheme or diagram) in relation to the specific criteria considered in the search carried out.

Finally, it is necessary to include a conclusion

Author Response

Dear Reviewer 2

Thank you very much for recognizing our work. Thank you for the opportunity to improve our manuscript with your valuable comments and recommendations.

This paper focuses on the treatment of dementia, specifically on the use of music therapy and its combination with pharmacological treatment as a possible strategy to improve functionality with respect to cognitive and non-cognitive dimensions of people diagnosed with dementia.   From a study and analysis of the published evidence regarding the effectiveness of music therapy in people diagnosed with dementia, in relation to cognitive and non-cognitive outcomes.

Thank you very much for recognizing our work.

The objective of the work is clear, and the methodology and analysis allows to achieve it. However, there are aspects that could be improved:

Thank you for the opportunity to improve our manuscript with your valuable comments and recommendations.

  1. In the methodology, the umbrella review protocol is used Include a reference that supports and justifies this approach.

Thanks for the suggestion. We will clarify according to your recommendation.

In Line 102 to 108. “The umbrella review methodology examines the quality and synthesizes the findings of multiple reviews in a field of research [22-23]. This type of research has nine steps: 1. Justify the need for revision; 2. Design the study and record the review protocol; 3. Elaborate the research strategy and conduct the bibliographical research; 4. Identify the revisions included; 5. Report the methodological quality of the included reviews; 6. Present the results of the review, 7. Report the findings and summary of evidence; 8. Limitations of the review report; 9. Conclusions [23], see table 1”

  1. Type of participants. It would be convenient to justify the age range of the study. Why over 18 years old?

Thanks for the suggestion. We will clarify according to your recommendation.

In line 149 to 154 “Even though older adults constitute the main bulk of the population stricken by dementia, this study also seeks to include people with young-onset dementia, who correspond to people 64 or fewer years old. Therefore, we decided to have the adult population affected by dementia in this study [25].”

  1. For a better global and specific understanding of each activity developed, a diagram could be included that individualizes the macro stages of the methodology and the tasks developed in each one of them.

Thanks for the suggestion. We will clarify according to your recommendation.

In line 109 - We introduce table 1 with steps and their description.

  1. The results could include a summary of the most relevant results (scheme or diagram) in relation to the specific criteria considered in the search carried out.

Thanks for the suggestion. We will clarify. As it is a protocol, we still don't have results.

In line 239 to 245 - 3. Results. The results will be presented in the form of tables. If homogeneous results are found, a meta-analysis graph will be displayed. The remaining data will be synthesized narratively. We intend to find contributions to the understanding of music therapy, the types of modalities, under which conditions it was applied, main cognitive and non-cognitive benefits, and, the facilitating aspects, the difficulties and strategies to deal with them during the implementation of the music-therapy.

  1. Finally, it is necessary to include a conclusion

Thanks for the suggestion. We will clarify. We changed section 5 to "Conclusion and implications to clinical practice"

Round 2

Reviewer 2 Report

The authors have adequately implemented the questions posed!